# The Association between the First Cry and Clinical Outcomes in CDH Neonates: A Retrospective Study

**DOI:** 10.3390/children10071145

**Published:** 2023-06-30

**Authors:** Yuki Takeuchi, Akiyoshi Nomura, Masaya Yamoto, Satoko Ohfuji, Shunsuke Fujii, Seiji Yoshimoto, Toru Funakoshi, Masato Shinkai, Naoto Urushihara, Akiko Yokoi

**Affiliations:** 1Department of Pediatric Surgery, Kobe Children’s Hospital, Kobe 650-0047, Japan; 2Department of Pediatric Surgery, Shizuoka Children’s Hospital, Shizuoka 420-0953, Japan; 3Department of Public Health, Osaka Metropolitan University Graduate School of Medicine, Osaka 545-8585, Japan; 4Department of Pediatric Surgery, Kanagawa Children’s Medical Center, Yokohama 232-0066, Japan; 5Department of Neonatology, Kobe Children’s Hospital, Kobe 650-0047, Japan; 6Department of Obstetrics, Kobe Children’s Hospital, Kobe 650-0047, Japan

**Keywords:** congenital diaphragmatic hernia, first cry, fetal sedation, survival

## Abstract

Congenital diaphragmatic hernia (CDH) is a life-threatening condition characterized by the herniation of abdominal organs into the thorax, resulting in hypoplastic lungs and pulmonary hypertension. The impact of the first cry, a crucial event for lung transition during birth, on CDH patients remains unclear. This study investigated the impact of the first cry during birth on CDH patient survival, along with other prognosis factors. A multi-institutional retrospective study assessed CDH patient characteristics and survival rates by analyzing factors including the first cry, disease severity, birth weight, Apgar scores, oxygenation index (OI) and surgical closure. Among the CDH patients in the study, a positive first cry was linked to 100% survival, regardless of disease severity (*p* < 0.001). Notably, the presence of a positive first cry did not significantly affect survival rates in patients with worse prognostic factors, such as low birth weight (<2500 g), high CDH severity, low Apgar scores (1 min ≤ 4), high best OI within 24 h after birth (≥8), or those who underwent patch closure. Furthermore, no significant association was found between the first cry and the use of inhaled nitric oxide (iNO) or extracorporeal membrane oxygenation (ECMO). In conclusion, this study suggests that the first cry may not have a negative impact on the prognosis of CDH patients and could potentially have a positive effect.

## 1. Introduction

Despite significant advancements in prenatal diagnosis and neonatal treatment of congenital diaphragmatic hernia (CDH) over the past few decades, the management of severe CDH remains challenging [1]. Management protocols vary globally with different guidelines offered in North America [2,3], Europe [4] and Japan [5]. Immediate post-birth endotracheal intubation and gentle ventilation with deep sedation are often recommended as the initial ventilation approach for prenatally diagnosed CDH patients. This strategy is proposed to reduce the risk of pulmonary hypertension due to prolonged acidosis and hypoxia, which might result from delayed intubation. For prenatally diagnosed CDH patients, the general recommendation is for immediate endotracheal intubation after birth, combined with gentle ventilation and deep sedation. This approach aims to circumvent the risks of pulmonary hypertension triggered by prolonged acidosis and hypoxia, which can ensue from delayed intubation. However, these recommendations are largely based on expert opinion and differ in specifics across different guidelines [6]. For instance, European and Canadian guidelines advocate for immediate intubation, allowing for up to 10 min in some cases [7]. In contrast, American guidelines do not provide explicit directives regarding immediate intubation for CDH infants in the delivery room [8]. Japanese guidelines suggest immediate intubation, typically within 1–5 min, to prevent air insufflation into the gastrointestinal tract after birth. Nonetheless, exceptions are made in mild antenatally diagnosed cases [5].

Interestingly, despite the diversity in these guidelines, none directly addresses the importance of a newborn’s first cry—a critical cardio-respiratory transition event from fetus to newborn. This event significantly influences organ function and blood flow [9]. As a consequence of immediate intubation protocols, this first cry may be inhibited, potentially interfering with a critical physiological process.

Our study aims to bridge this gap in the current guidelines by exploring the potential impact of the first cry on the management outcomes of CDH patients. To this end, we conducted a retrospective review of CDH cases across three centers, comparing survival rates, as well as the necessity for inhaled nitric oxide (iNO) and extracorporeal membrane oxygenation (ECMO), between patients who exhibited their first cry and those who did not. This study aims to shed light on whether the first cry could play a role in the delivery room management of CDH infants.

## 2. Materials and Methods

This retrospective study was approved by our hospital’s review board (R30-53). We conducted a retrospective analysis of the medical records of 81 infants diagnosed with isolated congenital diaphragmatic hernia (CDH) between January 2009 and September 2018 at three centers affiliated with the Japanese Congenital Diaphragmatic Hernia Study Group (JCDHSG). Infants with life-threatening severe associated anomalies, such as severe congenital heart disease or chromosomal anomalies, were excluded from the study. Patient management followed the guidelines of the JCDHSG [5].

In our study, the severity of CDH was classified using the observed/expected lung-to-head ratio (o/e LHR). This ultrasound measurement compares the area of the lung contralateral to the defect with the head circumference of the fetus. As lung size and head circumference evolve at different rates during pregnancy, a proportion of the observed (measured) over the expected (in normal fetus of the same gestational age) ratio (O/E LHR) provides a more accurate gauge of the fetus’s lung size relative to its gestational age [10]. The presence of a herniated liver is another key determinant in assessing the severity of CDH as its occurrence is positively correlated with mortality rates. The severity of CDH in fetuses is widely classified into four categories: extreme, severe, moderate and mild. This classification is based on a combination of liver herniation and the observed/expected lung-to-head ratio (o/e LHR). Fetuses with an o/e LHR less than 15% are categorized as having extreme CDH, while those with an o/e LHR ranging from 15% to 24.9% are classified as severe. Moderate CDH is defined as an o/e LHR of 25–34%, irrespective of liver position, or an o/e LHR of 35–44.9% in instances where the liver was herniated. Any case with an o/e LHR beyond these ranges is considered mild. Survival rates have been observed to increase as the severity decreases from severe to mild, with rates typically at 0% for extreme, 20% for severe, 30–60% for moderate and exceeding 75% for mild CDH [11,12]. For the purpose of further stratification in our study, extreme and severe categories were denoted as “High severity”, while moderate and mild were categorized as “Low severity”.

The mode of delivery was determined based on the assessment of prenatal severity and in accordance with the institutional policies for managing CDH. One center preferred cesarean section (CS) with maternal general anesthesia (CS with anesthesia) unless vaginal delivery was deemed safe for mild cases. The general anesthesia protocol at this center typically involved total intravenous anesthesia, which combined propofol and remifentanil, further supplemented with rocuronium bromide and fentanyl. Another center followed a policy of performing CS without maternal general anesthesia (CS without anesthesia) for all cases. The last center made the decision to either perform CS without anesthesia or opt for vaginal delivery, depending on the severity of the prenatal diagnosis.

Management protocols within the delivery room differed among the three centers participating in our study. However, immediate intubation of newborns, typically within the first minute post-birth, was a common practice across all centers. While the exact age at intubation was not specifically recorded, it is essential to underscore that all infants were intubated immediately after birth. Sedation protocols varied; some infants were indirectly sedated through maternal general anesthesia during delivery. In cases where maternal anesthesia wasn’t utilized, infants were initially intubated without sedation. After intubation, these infants were sedated with midazolam in combination with either fentanyl or morphine. The use of muscle-blocking agents was decided based on the severity of the CDH.

In our study, the presence of the first cry was assessed and recorded by experienced neonatologists at each center. The data was then collected retrospectively from patient charts. We also surveyed gestational age, birth weight, delivery method, the survival rate at 30 days and requirements for inhaled nitric oxide (iNO), extracorporeal membrane oxygenation (ECMO) and patch closure. The criteria for initiating treatment with iNO and ECMO varied across the centers. Generally, iNO was considered when SpO2 levels were not within the targeted pre-ductal range of 85–95% despite the use of 80–100% oxygen concentration. ECMO was initiated postoperatively when possible and typically considered when the alveolar-arterial oxygen gradient (AaDO2) exceeded 500 and the oxygenation index (OI) was over 0.4 for more than 4 h, despite maximal supportive treatments.

We assessed the association of several characteristics with the survival rate in patients with CDH. Each factor was classified as follows: gender (male or female), gestational age (≥37 weeks or <37 weeks), birth weight (≥2500 g or <2500 g), CDH severity (Low severity: mild and moderate CDH or High severity: severe and extreme CDH), Apgar scores at 1 min and 5 min (≤4 or ≥5 at both time points), best oxygenation index within 24 h after birth (Best OI)(<8 or ≥8), surgical approach (no surgery, laparotomy or thoracoscopy) and surgical closure method (Direct: direct surgical closure or Patch: closure with a patch).

Next, we examined the association between the presence (First cry positive) or absence (First cry negative) of the first cry and the 30-day survival rate, as well as the requirement for inhaled nitric oxide (iNO) and extracorporeal membrane oxygenation (ECMO), among the predictive factors for survival.

Statistical analyses, including chi-square or Fisher’s exact test, and Student’s *t*-test were performed to assess the significance of the observed associations. Data were expressed as mean ± standard deviation. A p-value threshold of less than 0.05 was considered statistically significant. All statistical analyses were conducted using JMP software (SAS Institute Inc., Cary, NC, USA).

## 3. Results

### 3.1. Delivery Modes and First Cry

Figure 1 illustrates the comparison of delivery modes between the Low severity and High severity groups. In the Low severity group, 23 patients had vaginal delivery, compared to only 2 patients in the High severity group. CS without anesthesia was chosen by 19 patients in the Low severity group and 13 patients in the High severity group, while CS with anesthesia was performed in 20 patients in the Low severity group and 3 patients in the High severity group. The First cry status for each delivery mode is presented in Figure 2. Among patients who underwent vaginal delivery, 20 were First cry positive, and 6 were First cry negative. In the CS without anesthesia group, 16 patients were First cry positive, and 16 were First cry negative. In the CS with anesthesia group, one patient was First cry positive, and 22 were First cry negative.

### 3.2. Prenatal Assessment of CDH Severity and First Cry

Among the severity groups, the presence of first cry was observed in 33 patients in the Low severity group and 3 patients in the High severity group, whereas the absence of first cry was noted in 29 patients in the Low severity group and 15 patients in the High severity group (Figure 3). There was a significantly higher occurrence of the first cry in the Low severity group compared to the High severity group (*p* = 0.0071).

Based on these results, the absence of first cry could be attributed to three potential reasons. Firstly, patients in the negative First cry group may have been too ill or unstable to produce a cry. Secondly, patients may have been intubated and sedated before the occurrence of the first cry. Lastly, prenatal sedation administered through maternal general anesthesia could have contributed to the absence of first cry in some patients. Based on the collected data, specific individual reasons for the absence of first cry could not be determined.

### 3.3. Comparison of All CDH Patients with and without First Cry

In our sample of CDH patients, 37 had their first cry (positive) while 44 did not (negative). The severity of CDH was significantly lower in the First cry positive group compared to the negative group (*p* = 0.03). Also, vaginal delivery, higher gestational age, greater birth weight and higher Apgar scores at 1 and 5 min were more common in the First cry positive group, all with a p-value of less than 0.01. The best Oxygenation Index (OI) was significantly lower in the First cry positive group (*p* < 0.01) though there was no significant difference in the use of inhaled nitric oxide (iNO) between the groups (*p* = 0.66). The incidence of ECMO use was higher in the First cry negative group, but this was not statistically significant (*p* = 0.12). In terms of surgery, all First cry positive patients underwent surgery, and a significantly higher number had direct surgical closure (*p* = 0.01). Importantly, the 30-day survival rate was significantly higher in the First cry positive group (*p* < 0.01) (Table 1).

### 3.4. Survival and First Cry

We investigated the impact of the presence of the first cry on the survival of CDH patients and examined other factors that could affect their survival (Table 2).

All CDH patients who had a positive first cry were found to be alive, demonstrating a significant association between the presence of a first cry and patient survival (*p* = 0.0003).

Furthermore, we observed that patients with a birth weight of ≥2500 g had a better outcome compared to those with a birth weight < 2500 g (*p* = 0.0047). Similarly, patients with a lower severity of CDH (mild and moderate) had a significantly higher survival rate compared to those with a higher severity (severe and extreme) (*p* < 0.0001). In terms of Apgar scores, patients with an Apgar score of 1 min ≥ 5 had a better survival rate than those with a score ≤ 4 (*p* = 0.0245). We found no significant association between a 5-min Apgar score ≥ 5 and survival. Patients with Best OI < 8 had a significantly better outcome than those with Best OI ≥ 8 (*p* = 0.0002). Regarding the surgical approach, the choice of either laparotomy or thoracoscopy did not appear to have a significant impact on patient survival (*p* = 1.000). However, patients who underwent direct closure had a better outcome compared to those who received patch closure (*p* = 0.0073). In summary, higher birth weight, lower severity of CDH, higher Apgar scores 1 min, better Best OI and direct closure were associated with improved survivals.

The impact of the first cry on CDH patient survival was examined within different prognostic groups. Analysis of the data revealed the following results:In patients with lower birth weight (<2500 g) or higher birth weight (>2500 g), the presence of a positive first cry did not significantly affect survival rates.Similarly, no significant difference in survival was observed between the positive and negative First cry groups in both mild/moderate and severe/extreme CDH cases.Survival rates did not significantly differ between the positive and negative first cry groups for patients with an Apgar score ≤ 4 or an Apgar score ≥ 5.Additionally, there was no significant difference in survival between the positive and negative First cry groups in patients with Best OI < 8 or Best OI ≥ 8.

The presence or absence of the first cry did not significantly impact survival rates in patients who underwent different surgical closure approaches. Overall, our analysis did not find a significant association between the presence of the first cry and survival among CDH patients, regardless of various prognostic factors (Table 3).

### 3.5. Need for iNO or ECMO and First Cry

Similarly, our analysis did not reveal any significant association between the presence of the first cry and the usage of iNO or ECMO, as shown in Table 4 and Table 5.

## 4. Discussion

One of the key findings of our study is that the presence of a first cry was associated with 100% survival among CDH patients, irrespective of disease severity. This observation suggests that the first cry does not exert a negative impact, even in patients with worse prognostic factors. These findings challenge the notion that the first cry may have detrimental effects on CDH patients and might provide reassurance regarding the potential benefits associated with a positive first cry.

The first cry is important for ensuring proper pulmonary circulation in newborn babies. The fetus’s lung is filled with fluid. During labor, hormone-mediated sodium reabsorption and surfactant secretion occur. The large negative pressure of the lung generated by crying forces the lung fluid from the airways and into the distal airspace [11]. Expiratory breath holds during crying facilitate lung volume recruitment [12]. As a consequence of pulmonary aeration, pulmonary vascular resistance significantly decreases and ductus arteriosus left-to-right shunting increases despite a decrease in diameter in healthy term infants [13].

While literature specifically addressing the first cry’s role in CDH is limited, recent studies on neonatal respiratory physiology provide valuable insights [14]. These cries can induce asymmetric ventilation, primarily in the right lung, an area that is typically unaffected in CDH, which is most commonly on the left side. This ventilation can lead to significant changes in alveolar pressure that play a crucial role in clearing lung fluid. Furthermore, the “pendelluft” flow—a dynamic process involving intrapulmonary gas transfer during the respiratory cycle—can occur. The “pendelluft” phenomenon, combined with the expiratory “braking” during crying, encourages the redistribution of ventilation to the more dependent areas of the lung. This enhanced aeration in these regions, coupled with pulmonary vasodilation associated with tidal respiration, can improve oxygenation significantly. Therefore, allowing the first cry might potentially be beneficial for neonates with CDH by assisting in the establishment of functional residual capacity, optimizing lung compliance and potentially preventing ventilation-induced lung injury.

Te Pas et al. demonstrated that infants with CDH exhibit significantly larger tidal volumes during spontaneous breathing compared to manual ventilation at birth [7]. Spontaneous breathing or consciousness may lead to better inflation of the pulmonary alveoli than manual ventilation. Positive pressure inflation immediately after birth decreases surfactant production, resulting in inflation of the lung with focal overextension and atelectasis [11]. In addition, spontaneous breathing potentially reduces pulmonary vascular resistance more efficiently than does manual ventilation.

It is worth noting that, within our cohort of severe CDH patients, only three exhibited the first cry. This could be attributed to potential concerns regarding the adverse effects of spontaneous respiration. Historically, the management of CDH has seen a preference for fetal sedation, aimed at intentionally inhibiting the neonate’s first cry upon birth [15]. Certain specialized institutions routinely opt for fetal sedation in cases of severe CDH that have been prenatally diagnosed. Senzaki et al. have previously reported instances of patients developing severe persistent fetal circulation due to the substantial mediastinal shift resulting from gas-filled gastrointestinal tracts, observed immediately upon admission to the NICU. They proposed employing deep sedation through the administration of morphine to the mother to inhibit spontaneous respiration, essentially “crying” [16]. However, Terui et al. noted that the group of patients who received fetal sedation demonstrated higher oxygenation indices compared to those who did not receive such treatment [17]. Current clinical guidelines for CDH patients in Japan widely accept the practice of “immediate” intubation after birth, typically within 5 min, without the administration of intravenous sedation (personal communication).

In our multicenter investigation, we found considerable variation across the three participating centers regarding the chosen mode of delivery for CDH patients. This variation is influenced by institutional protocols, clinical judgment and individual patient circumstances. Despite these differences, a universally agreed goal is shared by all centers in the management of neonates with CDH in the delivery room. This shared objective is the achievement of gentle ventilation, coupled with permissive hypercapnia, to prevent ventilation-induced lung injury and hypoxia, thereby protecting the newborn’s lung health. The noticeable disparity in the management approach underscores the relevance of our study. By comparing the outcomes from varying strategies, our findings may contribute to the development of future guidelines and standardize practices across different centers, potentially enhancing patient outcomes with CDH.

It is noteworthy that the controversy regarding the delivery room management of CDH is not confined to our institutions but is a global issue. A recent report from the Congenital Diaphragmatic Hernia Study Group indicates that 67% of North American clinical guidelines addressing CDH discuss the initial use of sedative medication. Among these, 70.4% advise against using paralytics for sedation [18]. Given this context, there is a clear need for further assessment of the role of fetal sedation in the management of CDH. Horn-Oudshoorn et al. have proposed a spontaneous breathing approach for patients with mild CDH [19]. While the prevailing strategy for managing CDH in the delivery room advocates immediate intubation to prevent gastric and bowel insufflation due to the infant’s respiratory efforts [20], an ongoing randomized trial is examining the option of mechanically ventilating the infant prior to cutting the umbilical cord [21].

Our study has several limitations that should be considered when interpreting the results. Firstly, although multiple centers were involved, the sample size was small, and the retrospective nature of the study introduces inherent limitations in data collection and potential biases. Secondly, the definition and assessment of the first cry varied across the participating centers, which may impact the interpretation of its effects on the cardio-respiratory system at birth. The strength and duration of the first cry could potentially influence outcomes differently. Thirdly, while we examined the presence of the first cry, we could not differentiate the reasons for its absence, such as the severity of the disease, immediate resuscitation efforts, or potential sedation induced by maternal general anesthesia during delivery. These factors may have influenced the observed associations and should be considered when interpreting the results.

Despite these limitations, our study represents the first multi-institutional investigation into the impact of the first cry on CDH management. This unique approach allowed us to gather data from different centers, enhancing the generalizability of the findings. However, future research with larger sample sizes and prospective designs is warranted to further validate and expand upon our findings.

In conclusion, our study suggests that the presence of the first cry may not have an adverse effect on the prognosis of CDH patients and could potentially have a positive impact. However, given the limitations of our study and the need for further research, it is important to approach management decisions for CDH patients on an individualized and tailored basis. Future investigations are warranted to gain a better understanding of the role of the first cry and to further evaluate the necessity of perinatal sedation in the management of CDH patients.

## Figures and Tables

**Figure 1 children-10-01145-f001:**
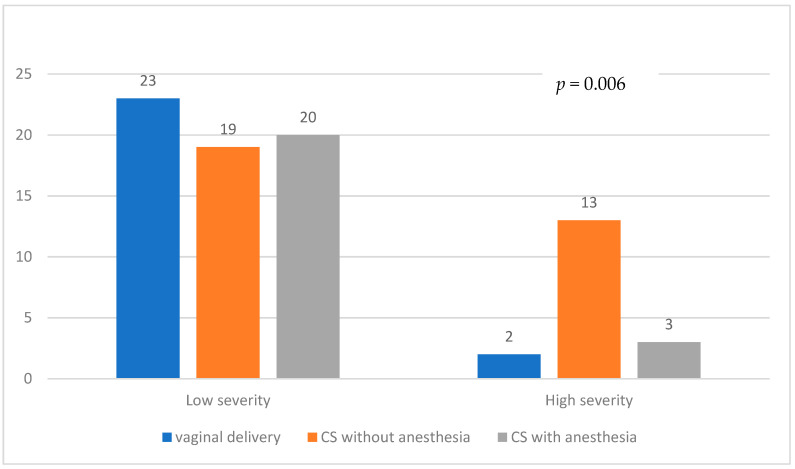
Modes of delivery and CDH severity. CDH Severity—Low Severity including mild (observed-to-expected lung-to-head ratio [o/e LHR] 36–45% and liver down or o/e LHR ≥ 46%) and moderate (o/e LHR 26–35% or o/e LHR 36–45% with intrathoracic liver position), High Severity including severe (o/e LHR 15–25%) and extreme (o/e LHR < 15%).

**Figure 2 children-10-01145-f002:**
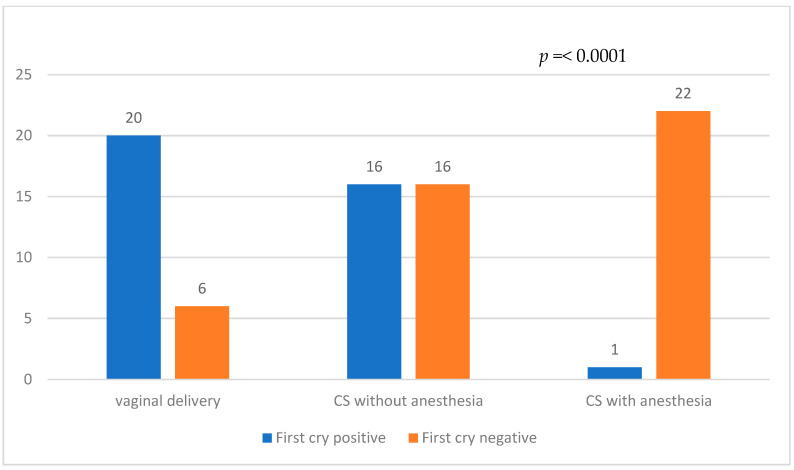
Comparison of First cry status among different delivery modes.

**Figure 3 children-10-01145-f003:**
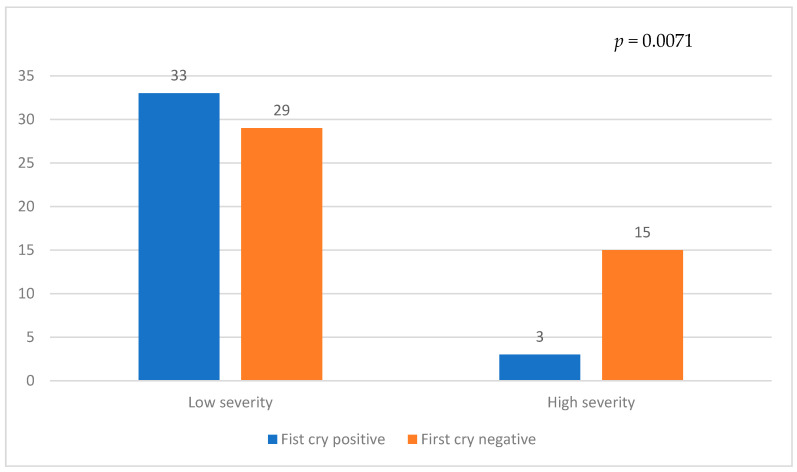
Comparison of First cry status with severity in CDH.

**Table 1 children-10-01145-t001:** Demographics on first cry in CDH patients.

		First Cry Positive (*n* = 37)	First Cry Negative (*n* = 44)	*p*-Value
CDH severity	mild	22 (61%)	16 (36%)	0.03
	moderate	11 (31%)	13 (30%)	
	severe	3 (8%)	12 (27%)	
	extreme	0	3 (7%)	
Gender	male	16 (43%)	23 (52%)	0.42
Delivery mode	vaginal	20 (54%)	6 (14%)	<0.01
Gestational age	median (range) weeks	38 (33–41)	37 (32–40)	<0.01
Birth weight	median (range) g	2987 (2054–3782)	2554 (1601–3340)	<0.01
Apgar 1 min	median (range)	6 (2–8)	2 (1–5)	<0.01
Apgar 5 min	median (range)	8 (3–9)	4 (1–8)	<0.01
Best OI	median (range)	3.6 (1.7–35.0)	7.3 (1.71–77.9)	<0.01
iNO		25 (68%)	31 (71%)	0.66
ECMO		0	4 (9%)	0.12
Age at surgery	median (range) day	2 (0–5)	1 (0–7)	0.08
Surgical approach	No surgery	0	6 (14%)	0.04
	laparotomy	32 (86%)	34 (77%)	
	thoracotomy	5 (14%)	4 (9%)	
Surgical closure	direct	31 (84%)	22 (58%)	0.01
	patch	6 (16%)	16 (42%)	
30 day survival		37 (100%)	32 (73%)	<0.01

**Table 2 children-10-01145-t002:** Comparison of characteristics in CDH survivors vs. non-survivors.

		Alive *n* = 69	Death *n* = 12	*p*-Value
First cry	positive	37	0	0.0003
	negative	32	12	
Gender	male	33	6	1.000
female		36	6	
Gestational age	≥37 weeks	64	9	0.0915
<37 weeks		5	3	
Birth weight	≥2500	53	4	0.0047
<2500		16	8	
CDH severity	Low (Mild and moderate)	59	3	<0.0001
High (Severe and extreme)		9	9	
Apgar score 1 min	≤4	39	11	0.0245
≥5		30	1	
Apgar score 5 min	≤4	27	8	0.1139
≥5		42	4	
Best OI	<8	49	1	0.0002
≥8		18	9	
Surgery (*n* = 75) approach	laparotomy	60	6	1.000
thoracoscopy		9	0	
closure	direct	52	1	0.0073
patch		17	5	

**Table 3 children-10-01145-t003:** Survival comparison based on first cry in CDH patients.

		First Cry Positive	First Cry Negative	*p*-Value
Birth weight	<2500 (*n* = 24)	5/5 (100%)	11/19 (57.89%)	0.1304
>2500 (*n* = 57)		32/32 (100%)	21/25 (84%)	0.032
CDH severity				
Low(Mild and moderate) (*n* = 62)		33/33 (100%)	26/29 (89.66%)	0.0966
High (Severe and extreme) (*n* = 18)		3/3 (100%)	6/15 (40%)	0.2059
Apgar score 1 min	≤4 (*n* = 50)	8/8 (100%)	31/42 (73.81%)	0.1737
≥5 (*n* = 31)		29/29 (100%)	1/2 (50.0%)	0.0645
Best OI	<8 (*n* = 50)	28/28 (100%)	21/22 (95.45%)	0.44
≥8 (*n* = 27)		7/7 (100%)	11/20 (55%)	0.0593
Surgical closure	direct (*n* = 53)	31/31 (100%)	21/22 (95.45%)	0.4151
patch (*n* = 22)		6/6 (100%)	11/16 (68.75%)	0.2663

**Table 4 children-10-01145-t004:** Usage of iNO comparison based on first cry in CDH patients.

		First Cry Positive	First Cry Negative	*p*-Value
Birth weight	<2500 (*n* = 23)	5/5 (100)	15/18 (83.33)	1
>2500 (*n* = 57)		20/32 (62.5)	16/25 (64)	1
CDH severity				
Low (Mild and moderate) (*n* = 62)		22/33 (66.67)	17/29 (58.62)	0.6019
High (Severe and extreme) (*n* = 17)		3/3 (100)	14/14 (100)	n.a.
Apgar score 1 min	≤4 (*n* = 49)	8/8 (100)	29/41 (70.73)	0.1732
≥5 (*n* = 31)		17/29 (58.62)	2/2 (100)	0.5097
Best OI	<8 (*n* = 50)	18/28 (64.29)	12/22 (54.44)	0.5669
≥8 (*n* = 27)		7/7 (100)	18/20 (90)	1
Surgical closure	direct (*n* = 53)	20/31 (64.52)	11/22 (50.00)	0.3976
patch (*n* = 22)		6/6 (83.33)	15/16 (93.75)	0.4805

**Table 5 children-10-01145-t005:** Usage of ECMO comparison based on first cry in CDH patients.

		First Cry Positive	First Cry Negative	*p*-Value
Birth weight	<2500 (*n* = 23)	0/5 (0)	1/18 (5.56)	1
>2500 (*n* = 57)		0/32 (0)	3/25 (12.0)	0.0786
CDH severity				
Low (Mild and moderate) (*n* = 62)		0/33 (0)	2/29 (6.90)	0.2147
High (Severe and extreme) (*n* = 17)		0/3 (0)	2/14 (14.29)	1
Apgar score 1 min	≤4 (*n* = 49)	0/8 (0)	4/41 (9.76)	1
≥5 (*n* = 31)		0/29 (0)	0/2 (0)	n.a.
Best OI	<8 (*n* = 50)	0/28 (0)	0/22 (0)	n.a.
≥8 (*n* = 27)		0/7 (0)	3/20 (15)	0.5453
Surgical closure	direct (*n* = 53)	0/31 (0)	0/22 (9)	n.a.
patch (*n* = 22)		0/6 (83.33)	4/16 (25.0)	0.5407

## Data Availability

Data are unavailable due to privacy or ethical restrictions.

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
