# Peer review of "The Association between the First Cry and Clinical Outcomes in CDH Neonates: A Retrospective Study"

_children, 2023, doi:10.3390/children10071145_

Round 1

Reviewer 1 Report

Overall, your work is commendable for its contribution to the understanding of how the first cry might influence the prognosis of neonates diagnosed with congenital diaphragmatic hernia (CDH). However, there are several points that need to be addressed to improve the manuscript:

The introduction is clear and well written. However, a concise overview of the existing literature regarding CDH management and the role of the first cry would further strengthen the context of the study.

Materials and Methods

a. There needs to be a clearer explanation of how the first cry was assessed. It is important to understand whether there were standardized criteria across the centers involved in the study.

b. The definition of severity groups should be clarified for ease of understanding. A brief explanation about observed/expected lung-to-head ratio (o/e LHR) and its significance in diagnosing CDH would be helpful for non-specialist readers.

c. It is unclear why the decision regarding the mode of delivery varied so much between the centers. Please provide more details about the factors influencing this decision-making process.

Discusion section can also be improved. The discussion would benefit from a deeper analysis of how the first cry might affect pulmonary circulation in neonates with CDH, given that it is a significant focus of your study. Further, more detailed comparison with previous literature of first cry in CDH needs to be presented. 

English language is adequate. Please proofread given typographical errors. 

Author Response

We deeply appreciate your insightful feedback on our manuscript. Please find below our responses to your valuable comments.

Materials and Methods

a. There needs to be a clearer explanation of how the first cry was assessed. It is important to understand whether there were standardized criteria across the centers involved in the study.

We appreciate your input and agree that a more explicit explanation regarding the assessment of the first cry is necessary. In our study, the presence of the first cry was assessed and recorded by experienced neonatologists at each center. The data was then collected retrospectively from patient charts. Given the nature of the multicenter study, the definition and assessment of the first cry varied across the participating centers, which is a limitation we acknowledge in the manuscript. Future studies should consider standardizing the definition and assessment of the first cry to provide more comparable data."

b.The definition of severity groups should be clarified for ease of understanding. A brief explanation about observed/expected lung-to-head ratio (o/e LHR) and its significance in diagnosing CDH would be helpful for non-specialist readers.

We thank you for the valuable suggestion. We added an explanation of severity.

c. It is unclear why the decision regarding the mode of delivery varied so much between the centers. Please provide more details about the factors influencing this decision-making process.

 We appreciate your observation on the variability of the decision-making process for the delivery mode in different centers. It is indeed true that our three participating centers exhibit substantial variability in delivery room management decisions, including delivery mode. This variability underscores the importance of our study in exploring the potential implications of these different strategies. However, it is essential to note that despite these differences, all centers share a common objective in the delivery room management of CDH infants. The primary goal is to achieve gentle ventilation and permissive hypercapnia, which prevents ventilation-induced lung injury and hypoxia. This shared aim unifies our approach to managing CDH across all participating centers.

Discusion section can also be improved. The discussion would benefit from a deeper analysis of how the first cry might affect pulmonary circulation in neonates with CDH, given that it is a significant focus of your study. Further, more detailed comparison with previous literature of first cry in CDH needs to be presented. 

We appreciate the reviewer's feedback regarding the depth of our discussion on the first cry's impact on pulmonary circulation in neonates with CDH. Though we could not locate literature directly addressing the first cry in CDH, we have found a valuable resource discussing the physiological implications of the first cry and tidal breathing following it.

This resource outlines how the first few cries result in asymmetrical ventilation of the lung, increased alveolar pressure aiding in lung liquid clearance, and improved aeration of dependent lung regions. It also discusses how tidal respiration leads to pulmonary vasodilation, thereby improving oxygenation. These processes may significantly impact pulmonary circulation in neonates with CDH, and we plan to incorporate these insights into our discussion. We apologize for any oversight in our previous manuscript and will ensure a more detailed comparison with relevant literature in our revised discussion section.

Reviewer 2 Report

The submitted article by Takeuchi & al investigated the impact of the first cry after birth on CDH newborn infants. The authors conducted a retrospective study at three Japanese centers and compared survival rates and the need for iNO and ECMO between infants who cried and those who did not.

The article is well-written and easy to follow, although some minor editing of the English language is required. However, the paper would be further strengthened by addressing the following major and minor issues:

1.       In the introduction, it is important to present the clinical relevance of this study.

2.       What is your local protocol guide for the management of infants in the delivery room? When do you intubate infants? What sedation do you use? It seems also interesting ton know the median age at intubation and the surgical closure.

3.       Which treatments do you use for maternal anesthesia?

4.       What are your criteria for initiating treatment with iNO and ECMO?

5.       It is important to analyze the severity of CDH, excluding infants born by cesarean with anesthesia.

6.       The comparison of Apgar at 5 mn are missing in your tables.

The article is well written and easy to follow even a minor editing of english language is required.

Author Response

We deeply appreciate your insightful feedback on our manuscript. Please find below our responses to your valuable comments.

  1. In the introduction, it is important to present the clinical relevance of this study.

Thank you for your valuable suggestion. We have now incorporated a more detailed explanation of the clinical relevance of our study in the introduction section. Specifically, we have elaborated on the management guidelines' variability for CDH across various geographical regions and discussed the potentially crucial role of the newborn's first cry. We hope this enhances the introduction's comprehensibility and outlines the significance of our research more effectively.

  1. What is your local protocol guide for the management of infants in the delivery room? When do you intubate infants? What sedation do you use? It seems also interesting ton know the median age at intubation and the surgical closure.

We appreciate your inquiry regarding our local protocols for managing infants with CDH in the delivery room. It is important to note that each of the three centers involved in our study has its own specific protocol for delivery room management.

In all three centers, immediate intubation of the newborns is a common practice. Typically, this procedure is performed within the first minute after birth. Sedation at the time of intubation varies. Some infants are indirectly sedated via maternal general anesthesia during delivery. However, for those who are not, intubation is performed without initial sedation, and subsequently, these infants are sedated using agents such as midazolam and fentanyl or morphine. The use of muscle-blocking agents is contingent on the severity of the CDH.

While we didn't specifically record the exact age at intubation, it is important to note that intubation was performed immediately after birth in all cases, typically within the first minute. Regarding the age at surgical closure, we have included this information in our demographics section. The median age at surgical closure was 1 day in the first cry (FC) negative group and 2 days in the FC positive group (p=0.08). We hope that this additional information clarifies the practices at our centers

  1. Which treatments do you use for maternal anesthesia?

We appreciate your query regarding our practices for maternal anesthesia. In our centers, we primarily utilize total intravenous anesthesia. This consists of a combination of propofol and remifentanil, augmented with rocuronium bromide and fentanyl.

  1. What are your criteria for initiating treatment with iNO and ECMO?

  We appreciate your interest in our criteria for initiating treatment with inhaled nitric oxide (iNO) and extracorporeal membrane oxygenation (ECMO). Please note that these protocols differ slightly among our three participating centers. Generally, we introduce iNO when conventional oxygen therapy, applied at a concentration of 80-100%, fails to elevate the pre-ductal SpO2 levels to the target range of 85–95%. This is in line with our goal of ensuring adequate oxygenation while mitigating potential oxygen toxicity. As for the initiation of ECMO, it's typically employed postoperatively when feasible, and the critical indicators are an alveolar-arterial oxygen gradient (AaDO2) exceeding 500 and an oxygenation index (OI) above 0.4, sustained for more than four hours despite maximum supportive treatments.

  1. It is important to analyze the severity of CDH, excluding infants born by cesarean with anesthesia.

We appreciate the reviewer's insight and concern about the possible confounding effect of cesarean section with anesthesia. However, it is precisely this group of infants - those born by cesarean section under anesthesia, and thus potentially missing the opportunity to cry immediately after birth - that we wish to include in our study. As the initial cry could potentially influence the prognosis of CDH, the inclusion of this cohort is essential for us to comprehensively examine the effect of the first cry on CDH outcomes. The severity of CDH, as determined prenatally via ultrasound, is independent of the delivery mode or use of anesthesia, so this should not bias our results.

  1. The comparison of Apgar at 5 mn are missing in your tables.

 We appreciate your suggestion to include a comparison of Apgar scores at 5 minutes in our tables. However, during our preliminary analysis (as shown in Table 2), we found that an Apgar score at 5 minutes of over 5 was not associated with survival. As our focus was on factors significantly associated with survival, we opted not to include Apgar scores at 5 minutes in our further survival comparison analysis based on the presence of the first cry. Nevertheless, we recognize the importance of transparency and will clarify this decision-making process in the manuscript.

Round 2

Reviewer 1 Report

comments have been addressed. thank you

Author Response

Thank you for taking the time to review our manuscript and for acknowledging that we have adequately addressed your previous comments. 

Reviewer 2 Report

Congratulations for this study. There is some keyboard typos that need to be corrected like delivery mode "virginal"...

Author Response

We have corrected the typo you identified ("virginal" to "vaginal"). We have also thoroughly reviewed the manuscript again for any additional typographical errors, but could not find any. 

f you have noticed any other errors, we would appreciate it if you could specify them for us.

Again, we appreciate your attention to detail in helping us improve our manuscript.